# Prevalence and factors associated with anaemia among pregnant women attending reproductive and child health clinics in Mbeya region, Tanzania

Fatma Abdallah[1], Sauli E. John[1], Adam Hancy[1], Heavenlight A. Paulo[2], Abraham Sanga[3], Ramadhan Noor[3], Fatoumata Lankoande[3], Kudakwashe Chimanya[3], Ray M. Masumo[1]*, Germana H. Leyna[1,2]

1 Tanzania Food and Nutrition Centre, Dar es Salaam, Tanzania, 2 Department of Epidemiology and Biostatistics, Muhimbili University of Health and Allied Sciences, MUHAS, Dar es Salaam, Tanzania, 3 The United Nations Children's Fund (UNICEF) Tanzania, Dar es Salaam, Tanzania

* rmasumo@yahoo.com

**Data Availability Statement:** All datasets underlying this study are freely available at the public repository https://osf.io/7ysb9/.

## Abstract

Anaemia is a global public health issue, disproportionately affecting vulnerable populations such as pregnant women. The aim of this study was to assess the prevalence of anaemia and to identify factors associated with the condition among pregnant women attending antenatal clinics in the Mbeya Region of Tanzania. A cross sectional study was conducted with 420 pregnant women (<28 weeks of gestation) attending antenatal visits in the 7 districts of the Mbeya Region. A structured questionnaire was used to collect demographic information and eating habits using a 24hours dietary recall. A blood sample was collected and tested for hemoglobin content using the HemoCue 201+. Multivariate analysis was performed using standard logistic regression to explore the association between anaemia status with socio-demographic, reproductive and nutritional factors.

Overall prevalence of anaemia in pregnant women was 25.5%. Out of 107 pregnant women diagnosed with anaemia and, sixty six had mild anaemia. In a multivariate logistical regression analysis anaemic women was associated with pregnant women coming from lower socio-economic status [adjusted OR = 2.40, 95%CI (1.05, 5.48)]. Moreover, anaemia was less associated with pregnant women who were living in Mbeya district council [adjusted OR = 0.28, 95%CI (0.11, 0.72)], consume at least once a day dark green leafy vegetables [adjusted OR = 0.53, 95% CI (0.30, 0.94)], and vegetable liquid cooking oil [adjusted OR = 0.56, 95% CI (0.34, 0.98)]. The prevalence of anaemia among the pregnant women falls in the category of moderate public health problem according to the WHO classification. Low socio-economic status, consumption of green leafy vegetables and vegetable liquid cooking oil were significantly and independently associated with anaemia during pregnancy. Thus, special attention should be given to pregnant women who are in lower socio-economic status and those not consuming vegetables. Interventions that integrate health and nutrition education in reproductive and child health clinics are needed to combat anaemia.

**Funding:** This research received grants from the United Nations Children's Fund (UNICEF) – Tanzania to GHL. The funders had no role in study design, data collection and analysis, decision to publish, or preparation of the manuscript.

**Competing interests:** The authors have declared that no competing interests exist.

## Background

Iron deficiency anaemia in pregnancy is a serious public health problem in many developing countries, associated with maternal and perinatal mortality, premature delivery, low birth weight, and other adverse outcomes [1]. The World Health Organization (WHO) defines anaemia in pregnancy as a hemoglobin concentration below 11g/dL or hematocrit of <33% [2]. Globally more than half of all pregnant women have hemoglobin levels indicative of anaemia [3]. In industrialized countries, the prevalence of anaemia among pregnant women is 15% and in developing countries, prevalence ranges between 33% and 75% [4]. According to the WHO, anaemia is recognized as a public health problem if prevalence is 5.0% or higher [2, 5]. Prevalence of anaemia of equal or more than 40% in a population is classified as a severe public health problem [2, 5].

Previous studies indicate that the prevalence of anaemia is highest among pregnant women in Sub-Saharan Africa (57%) and Southeast Asia (48%) [2, 5]. The lowest prevalence (24.1%) was found among pregnant women in South America [2, 5]. The recent published study on Demographic and Health Surveys spanning from 2000 to 2018 included the data sets of 37,623 pregnant women reported that prevalence of anaemia among pregnant women in Sub-Saharan Africa has not improved since 2000 [6]. Data from the Tanzania Demographic and Health Survey (TDHS) between 2005 and 2010 suggests that the prevalence of anaemia in pregnant women decreased from 58% to 53% [5, 7, 8] but increased to 57% in 2015/16 [8]. Other research conducted in Tanzania has reported a higher prevalence of anaemia. For instance, in 2009, Kidanto and colleagues reported a prevalence of 68% in the Dar es Salaam Region and in 2011, Msuya and colleagues reported a prevalence of 47% in Moshi [9, 10].

Evidence suggests that the causes of anaemia are multifactorial, including micronutrient deficiencies of iron, folate, and vitamins A and B12, parasitic infections such as malaria and hookworm, and chronic infections [11, 12]. Contributions of each of the factors to anaemia in pregnancy vary due to geographical locations, dietary practices, and seasons [13]. Studies from the Sub-Saharan Africa region have reported that inadequate intake of dietary iron is the leading cause of anaemia among pregnant women [14]. During pregnancy, there is a marked increase in the minimum adult requirement by almost 2–3 folds for iron and 10–20 folds for folate [15]. Requirements also vary by gestational stage; a pregnant woman of 55kg requires approximately 0.8mg of iron in the first trimester, 4–5mg of iron during the second trimester and above 6mg of iron in the third trimester [16]. However, published literatures in the Sub-Saharan Africa countries exploring socio- economic, demographic and behavioral factors that predispose to anaemia in pregnancy remained scarce [17]. Previous studies have significantly improved our understanding on the impact of preventing maternal anaemia and reduce maternal mortality by about 20%, which is a significant reduction, and important given the fact that the maternal mortality rate in Tanzania is high [18, 19].

The consequences of anaemia are connected with decreased in personal productivity and in turn have substantial economic losses to the country [17, 20]. The annual productivity loss caused by anaemia is approximately USD 3.64 per capita, which is about 0.81 percent of Gross Domestic Product (GDP) of ten developing countries [20]. In Tanzania, the government has put in place several policies and interventions including anaemia screening, iron and folic acid supplementation, deworming, intermittent prophylaxis treatment for malaria using sulfadoxine pyrimethamine (SP), provision of free mosquito treated nets and, nutrition education during antenatal visits [19]. However, there is no current information on the burden and factors associated with anaemia during pregnancy following such interventions. To improve our understanding of the determinants and prevalence of anaemia, the present study aims to assess

the prevalence of anaemia and to identify factors associated with anaemia among pregnant women attending antenatal clinics (ANC) in the Mbeya Region of Tanzania.

## Methods

### Ethics statement

Ethical clearance was obtained from the National Institute for Medical Research (NIMR), Ethical Review Committee with Reference Number SZECH-2439/R.A/V.1/49. Written and verbal consent was obtained from all study participants. The risks and benefits of the study were clearly explained to the participants before signing the consent. Also participants were informed of their rights to refuse to participate in the study or withdraw from it at any time, without consequences. All procedures followed were per the ethical standards of the Helsinki Declaration of 1975 including the confidentiality.

### Study setting and design

A cross sectional survey of 420 pregnant women (gestation age below 28 weeks) registered at the reproductive and child health clinics, aged between 15 and 49 years from the seven districts of Mbeya Region was conducted. The study was carried out from September 2020 to October 2020. The Mbeya region has a population of 2,707,410 [21], and in 2020 the region had 318 health facilities of which 17 were hospitals, 23 health centers, and 278 dispensaries, with 251 of the health facilities (both government, private and faith-based organizations) providing reproductive and child health services. This study was conducted in 42 Reproductive and Child Health (RCH) Clinics across the seven districts of Mbeya region. The allocation of RCH per district for was based to probability proportional to size: Mbeya District Council (n = 11); Chunya District Council (n = 4); Mbeya City (n = 3); Mbarali District Council (n = 8); Kyela District Council (n = 6); Rungwe District Council (n = 7) and, Busekelo District Council (n = 4). The selected RCH clinics are estimated to provide services to approximately 1036 pregnant women.

### Study population

All pregnant women who attended RCH clinics within their first and second trimesters (less than 28 weeks of gestation) were invited to participate in the study. A total of 574 pregnant women were eligible, and 420 were selected to take part, as per the calculated sample size. Pregnant women who refused to consent and those who were unable to communicate due to illness or taking medication were excluded from the study.

### Sample size and sampling procedure

The prevalence of anaemia among women of reproductive age reported in the Tanzania Demographic and Health Survey (TDHS) in 2015 was 45%. Based on this figure and the population of pregnant women in this region, a sample size of 574 was calculated using the Lwanga and Lemeshow formula [22] with: margin error of 5%; confidence level of 95%; design effect of 1.5 and; an additional of 10% to account for non-response. The obtained sample size of 420 was considered satisfactory assuming intracluster correlation coefficient (ICC) of 0.10 and, a power of 80%.

The sampling procedure involved two steps: First, a list of 251 government, private and faith-based health facilities providing RCH services in Mbeya region was obtained and used in a random selection of the health facilities to be involved in the study from each district. Given the sampling frame of health facilities in Mbeya, probability proportional to size was

performed to allocate the number of facilities per District for inclusion in the survey. Out of 251 health facilities that offer RCH services (eligibility criteria) in Mbeya, forty two facilities were randomly selected for the study. An additional two reserved clusters were included in the survey. Therefore, a total of 44 health facilities offering RCH services located in the Mbeya region were visited and surveyed.

The second step involved the selection of pregnant women for each selected health facility. An eligibility form was used to list all pregnant women attending ANC services in the selected health facility. The resulting list of pregnant mothers served as the sampling frame for the selection of participants who met the inclusion criteria. Systematic Random Sampling was then carried out by using the list of mothers to randomly select required pregnant women for each facility to participate in the survey based on probability and proportion to size sampling for the specific facility.

## Data collection

**Dietary quality assessment by the Prime Diet Quality Score.** The Prime Diet Quality Score (PDQS) was recent developed using a modified Prime Screen questionnaire as a way to characterize diet quality globally [23]. PDQS contains 20 food groups; 13 are healthy food and, 7 are unhealthy food. PDQS assessed using a 24-hour recalls, which reflected the feeding practice from the previous morning to the morning of the interview. A standard structured questionnaire of PDQS constructed in English was translated into Kiswahili, the main language in Tanzania, spoken proficiently by almost 95% of the population. The questionnaire was translated; from English into Kiswahili by bi-lingual Kiswahili/English, and then back translated to English by independent translators. Project staff in the field reviewed for semantic, experiential and conceptual equivalence to the original version. Sensitivity to culture and selection of appropriate words were considered. The structured questionnaire was piloted to a separate group of women (not part of this study) to evaluate the quality of the translations in terms of comprehensibility, readability and relevance to assess face validity.

Pregnant women were asked 'from when you woke up yesterday till you woke up this morning did you consume the following food items: dark green leafy vegetables, cruciferous vegetables, dark orange vegetables and fruits, other vegetables, citrus fruits, other fruits, legumes, nuts and seeds, poultry, fish, whole grains, vegetable liquid oils, white roots and tubers, red meat as a main dish, processed meats, refined grains and baked products, sugar-sweetened beverages, fried foods away from home, sweets and ice cream, low-fat dairy?' Responses were given on a 5- point likert scale [23].

**Demographic and socio-economic factors.** All pregnant women who met inclusion criteria and provide a consent were invited to face to face interview guided by a structured coded questionnaire programmed into the Open Data Kit (ODK) and administered using Android tablets. The demographic and socio-economic variables considered were: age of pregnant mother, age of pregnancy, parity status, marital status, education level, occupation status, household assets, number of ANC visits, smoking and alcohol consumption.

**Anthropometry measurements.** Weight was measured to the nearest 0.1 kg with a battery-powered electronic scale (Seca, Hamburg, Germany) and height was measured to the nearest 0.1 cm with a height model recommended by UNICEF. Height was measured when the subject was not wearing shoes or a head covering. Mid Upper Arm circumference (MUAC) was assessed to pregnant women using MUAC tapes. All procedures were repeated to check for accuracy.

**Laboratory investigations.** In each health facility, a temporary laboratory was set for sample collection and field-testing. A trained nurse collected blood samples through vein puncture

from consented participants. The sample was collected in an EDTA vacutainer tube and was used to test for malaria via a rapid diagnostic test and hemoglobin levels using the HemoCue 201+.

## Data analysis

The data were analyzed using SPSS version 22. Frequency tables were used to summarize participant information. The dependent variable was anaemia status of pregnant women aged 15–49 years, and the independent variables were maternal factors (age, educational level, occupation, marital status, parity, household assets, geographical locations (councils), ANC visits, smoking and alcohol consumption), intake of folic acid and iron supplement, and dietary quality using PDQS. Univariate analysis was performed using chi-square and standard logistic regression with odds ratios (OR), and 95% Confidence intervals (CI) to explore the association between outcome variable and independent variables. Independent variables selected for the Multivariable logistic regression were included if they had a *P-value* equal to 0.05 in the univariate analysis.

Household wealth was also assessed as an indicator of socio-economic status according to a standard approach in equity analysis [24]. Durable household assets indicative of wealth (i.e. radio, television, telephone, refrigerator, bicycle, motorcycle etc.) were recorded as (1) "available and in working condition" or (0) "not available and/or not in working condition." These assets were analyzed using principal components analysis (PCA). The first component resulting from this analysis was used to categorize households into five approximate quartiles of wealth ranging from the 1$^{st}$ quartile (highest- richest) to the 5$^{th}$ quartile (lowest-poorest).

The 24-hour recalls reflected feeding practices from the previous morning to the morning of the interview and were used to assess the intake of the 20 food groups from the Prime Diet Quality Score (PDQS). The recalls were conducted by asking pregnant women 'from when you woke up yesterday till you woke up this morning did you consume the following food items (frequency of eating dark leafy green vegetables, whole grains, vegetable liquid oils, red meats and rrefined gains and baked goods). Responses were given on a 5- point Likert scale; 0 = never, 1 = once, 2 = twice, 3 = thrice and 4 = four or more. The scores were summarized and categorized into two groups i.e. 0 = not at all, 1 = at least once.

## Results

### Prevalence of anaemia in the study area

The overall prevalence of anaemia among pregnant women in Mbeya region Tanzania was 25.5%. As presented in Table 1, one hundred and seven pregnant women who were diagnosed with anaemia, sixty-six had haemoglobin of range between 10.0 to 10.9 g/dl (that representing a mild anaemia) and, only one had haemoglobin of less than 7.0g/dl that representing severe anaemia. Severely anaemic were referred to the nearby health facility for further investigation and treatment.

Table 1. Distribution of anaemia among pregnant women according to severity in Mbeya Tanzania (n = 107).

| Hb (range in g/dL) | Severity of anaemia | No. of pregnant women (*n* = 107) | Percentage |
|---|---|---|---|
| <7 | Severe | 1 | 0.93 |
| 7–10 | Moderate | 40 | 37.38 |
| 10–10.9 | Mild | 66 | 61.68 |

Hb: Hemoglobin

## Characteristics of the study population

Completed questionnaires and biochemical data from 420 pregnant women were gathered. Table 2 summarizes participant characteristics. Participants ranged in age from 15 to 49 years, with most (55%) aged between 20 and 29 years. Almost three-quarters (72%) had primary education, 65% were grand multipara and 57% were married. Most (74%) were in their second trimester (12–26) of pregnancy and about half visited antenatal clinics only two to three times. Out of the 420 respondents to the questionnaire, more than a half (51%) reported to take Fansidar (SP) during current pregnancy and only eighteen (4.2%) of pregnant women tested positive for malaria. Pregnant women who tested positive for malaria were referred to the nearby health facility for further investigations and treatment. Moreover, only 37% of the respondents received iron and folic acid supplements, 16.7% drink alcohol and, 4.3% are smoking.

Assessments of each of the 20 components of PDQS to investigate whether specific dietary components might explain any observed associations with anaemia among pregnant women in Mbeya. As presented in Table 2, majority of the participants (66%) consumed dark leafy green vegetables daily, whilst 26% consumed red meat at least once a day. In addition, 89% of pregnant women reported consuming vegetable liquid oils at least once a day.

## Univariate logistic regression: Anaemia with socio-demographic, reproductive and, nutrition characteristics

As shown in Table 3, pregnant women who attained secondary education or higher were less likely to be anaemic [uOR (unadjusted Odds Ratio) = 0.49, 95%CI (0.26, 0.92)]. Similarly, pregnant women from poor socio-economic status (household wealth index: lower and middle quintiles) were more likely to be anaemic [(uOR = 1.78, 95%CI (1.02, 3.09)] and, [uOR = 1.97, 95%CI (1.15, 3.39)] respectively. The prevalence of anaemia was significantly associated with the geographical locations (Councils): Mbeya district council [uOR = 0.23, 95%CI (0.09, 0.54)]; Busekelo district council [uOR = 0.29, 95%CI (0.09, 0.90)] and; Mbeya city council [uOR = 0.10, 95%CI (0.02, 0.50)]. Moreover, those pregnant women who received iron and folic acid supplement were found protective to anaemia [(uOR = 0.65 95% CI (0.41, 1.00)]. As also shown in Table 3, the univariate analysis of 20 components of PDQS, only two components were statistic significantly associated with anaemia: dark leafy green vegetables [uOR = 0.56, 95%CI (0.34, 0.91)] and, vegetable liquid oil [uOR = 0.50, 95%CI (0.26, 0.94)].

## Multivariate logistic regression: Anaemia with socio-demographic, reproductive health and nutrition characteristics among anaemic pregnant women to control cofounders

Education level, household wealth index, geographical location (councils), consumption of dark green leafy vegetables, vegetable liquid oils, and received iron and folic acid supplementation were selected for the multivariable model Table 4. The results showed that poor household wealth indices and, geographical location (Mbeya district council) were significantly associated with anaemia [aOR (adjusted Odds Ratio) = 2.40, 95%CI (1.05, 5.48)] and, [aOR = 0.28, 95% CI (0.11, 0.72)] respectively. Also, in terms of the nutritional habits of pregnant women, the frequency of eating of dark green leafy vegetables, [aOR = 0.53, 95% CI (0.30, 0.94)] and use of vegetable liquid oil for at least once a day, [aOR = 0.56, 95% CI (0.34, 0.98)] were significantly associated with anaemia.

**Table 2. The socio-demographic reproductive health, and nutrition characteristics among pregnant women attending Antenatal clinics in Mbeya, Tanzania (n = 420).**

| Variables | Category | % (n) |
|---|---|---|
| Age group (years) | 15–19 years | 19.5 (82) |
| | 20–24 years | 31.7 (133) |
| | 25–29 years | 23.6 (99) |
| | ≥30 years | 25.2 (106) |
| Marital Status | Married | 56.5 (238) |
| | Cohabit | 31.8 (133) |
| | Single/ Divorced | 11.7 (49) |
| Occupation | Formal employment | 3.6 (15) |
| | Self employed | 84.5 (355) |
| | Not employed | 11.9 (50) |
| Household wealth index | Highest quintile-richest | 20 (84) |
| | Richer | 20 (84) |
| | Middle | 20 (84) |
| | Poorer | 20 (84) |
| | Lowest quintile-poorest | 20 (84) |
| Education Status | No education | 8.1(34) |
| | Primary | 71.7 (301) |
| | Secondary and above | 20.2 (85) |
| Councils | Chunya district council | 10.7 (45) |
| | Mbeya district council | 23.0 (96) |
| | Mbarali district council | 22.1 (93) |
| | Kyela district council | 11.9 (50) |
| | Rungwe district council | 16.4 (69) |
| | Busekelo district council | 7.8 (33) |
| | Mbeya city | 8.1 (34) |
| Number of ANC care visit during this pregnancy | First visit | 38.8 (163) |
| | 2–3 Visits | 53.8 (226) |
| | than 3 visits | 7.4 (31) |
| MUAC | MUAC<23cm | 3.8 (16) |
| | MUAC> = 23 -<33cms | 91.2 (383) |
| | MUAC> = 33cms | 5 (21) |
| Trimester | Less than 12 weeks | 26.0 (109) |
| | 12–26 weeks | 74.0 (311) |
| Parity | Nulliparity | 1.0 (04) |

*(Continued)*

**Table 2.** (Continued)

| Variables | Category | % (n) |
|---|---|---|
| | Para 1–2 | 50.4 (212) |
| | Para 3–4 | 33.0 (139) |
| | Grand multipara | 15.4 (65) |
| Vaginal Bleeding | No | 95.5 (401) |
| | Yes | 4.5 (19) |
| Number of Abortion | None | 77.4 (325) |
| | One | 18.3 (77) |
| | Two or more | 4.3 (18) |
| Received Fansidar (SP) during this pregnancy | No | 48.6 (204) |
| | Yes | 51.4 (216) |
| History of Severe anaemia (previously diagnosed by a health worker) | No | 97.9 (411) |
| | Yes | 2.1 (9) |
| Malaria (Parasitemia) Status | Positive | 4.2 (18) |
| | Negative | 95.8 (402) |
| Received Iron/folic acid Supplementation | No | 63.1 (155) |
| | Yes | 36.9 (265) |
| Are you consuming alcohol such as beer, wine, spirits or local brews? | No | 83.3 (350) |
| | Yes | 16.7 (70) |
| Are you smoking? | No | 95.7 (402) |
| | Yes | 4.3 (18) |
| PDQS: **20 food groups** | | |
| *1. From yesterday morning to today morning did you eat dark leafy green vegetables* | Not at all | 33.8 (142) |
| | At least once | 66.2 (278) |
| *2. From yesterday morning to today morning did you eat cruciferous vegetables* | Not at all | 93.3 (392) |
| | At least once | 6.7 (28) |
| *3. From yesterday morning to today morning did you eat dark orange vegetables and fruits* | Not at all | 62.4 (262) |
| | At least once | 37.6 (158) |
| *4. From yesterday morning to today morning did you eat other vegetables* | Not at all | 63.1 (265) |
| | At least once | 36.9 (155) |
| *5. From yesterday morning to today morning did you eat whole citrus fruits* | Not at all | 92.9 (390) |
| | At least once | 7.1 (30) |

(*Continued*)

**Table 2.** (Continued)

| Variables | Category | % (n) |
|---|---|---|
| 6. From yesterday morning to today morning did you eat other whole fruits | Not at all | 73.1 (307) |
| | At least once | 26.9 (113) |
| 7. From yesterday morning to today morning did you eat legumes | Not at all | 61.4 (258) |
| | At least once | 38.6 (162) |
| 8. From yesterday morning to today morning did you eat nuts and seeds | Not at all | 65.5 (275) |
| | At least once | 34.5 (145) |
| 9. From yesterday morning to today morning did you eat poultry | Not at all | 92.4 (388) |
| | At least once | 7.6 (32) |
| 10. From yesterday morning to today morning did you eat fish | Not at all | 65.7 (276) |
| | At least once | 34.3 (144) |
| 11. From yesterday morning to today morning did you eat whole grains | Not at all | 76.9 (323) |
| | At least once | 23.1 (97) |
| 12. From yesterday morning to today morning did you eat vegetable liquid oils | Not at all | 11 (46) |
| | At least once | 89 (374) |
| 13. From yesterday morning to today morning did you eat white roots and tubers | Not at all | 44.8 (188) |
| | At least once | 55.2 (232) |
| 14. From yesterday morning to today morning did you eat red meats | Not at all | 73.8 (310) |
| | At least once | 26.2 (110) |
| 15. From yesterday morning to today morning did you eat processed meat | Not at all | 97.9 (411) |
| | At least once | 2.1 (9) |
| 16. From yesterday morning to today morning did you eat rrefined gains and baked goods | Not at all | 17.6 (74) |
| | At least once | 82.4 (346) |
| 17. From yesterday morning to today morning did you eat sugar sweetened beverages | Not at all | 59.3 (249) |
| | At least once | 40.7 (171) |
| 18. From yesterday morning to today morning did you eat fried food | Not at all | 82.6 (347) |
| | At least once | 17.4 (73) |
| 19. From yesterday morning to today morning did you eat sweets and ice cream | Not at all | 82.9 (348) |
| | At least once | 17.1 (72) |
| 20. From yesterday morning to today morning did you eat low fat diary | Not at all | 83.1 (349) |
| | At least once | 16.9 (71) |

**Table 3. Univariate logistic regression of anaemia status with socio-demographic, reproductive health, and nutrition characteristics among pregnant women attending Antenatal clinics in Mbeya, Tanzania (n = 420).**

| Variables | Category | Anaemic status | |
|---|---|---|---|
| | | % (n) | uOR (95% CI) |
| Age | 15–19 years | 25.6 (21) | 1 |
| | 20–24 years | 21.1 (28) | 1.29 (0.67, 2.46) |
| | 25–29 years | 32.3 (32) | 0.72 (0.37, 1.36) |
| | ≥30 years | 24.5 (26) | 1.05 (0.54, 2.05) |
| Household wealth index | Highest quintile- richest | 17.9 (15) | 1 |
| | Richer | 32.1 (27) | 1.44 (0.80, 2.59) |
| | Middle | 35.7 (30) | **1.97 (1.15, 3.39)** |
| | Poorer | 26.2 (22) | **1.77 (1.02, 3.09)** |
| | Lowest quintile- poorest | 15.5 (13) | 0.86 (0.43, 1.70) |
| Marital Status | Married | 23.1 (55) | 1 |
| | Cohabit | 27.8 (37) | 0.78 (0.48, 1.26) |
| | Single/ Divorced | 30.6 (15) | 0.68 (0.34, 1.34) |
| Occupation | Not employed | 20.0 (3) | 1 |
| | Self employed | 25.6 (91) | 0.71 (0.17, 2.92) |
| | Formal employment | 26.0 (13) | 0.98 (0.50, 1.94) |
| Education Status | No education | 38.2 (13) | 1 |
| | Primary | 25.9 (78) | 0.68 (0.42, 1.08) |
| | Secondary and above | 18.8 (16) | **0.498 (0.26, 0.92)** |
| Councils | Chunya district council | 62.2 (28) | 1 |
| | Mbeya district council | 87.6 (85) | **0.23 (0.09, 0.54)** |
| | Mbarali district council | 60.9 (56) | 1.05 (0.50, 2.20) |
| | Kyela district council | 66.0 (33) | 0.84 (0.36, 1.96) |
| | Rungwe district council | 73.5 (50) | 0.59 (0.26, 1.33) |
| | Busekelo district council | 84.8 (28) | **0.29 (0.09, 0.90)** |
| | Mbeya city | 93.9 (31) | **0.10 (0.02, 0.50)** |
| Number of ANC care visit during this pregnancy | First visit | 27.0 (44) | 1 |
| | 2–3 Visits | 25.7 (58) | 0.94 (0.67, 1.32) |
| | More than 3 visits | 16.1 (15) | 0.59 (0.25, 1.37) |
| MUAC | MUAC<23cm | 37.5 (6) | 1 |
| | MUAC> = 23 - <33cms | 24.8 (95) | 0.55 (0.19, 1.54) |
| | MUAC> = 33cms | 28.6 (6) | 0.66 (0.16, 2.66) |
| Trimester | Less than 12 weeks | 21.3 (33) | 1 |
| | 12–26 weeks | 26.9 (84) | 1.27 (0.84, 1.91) |
| Parity | Nulliparity | 0.0 (0.0) | |
| | Para 1–2 | 22.9 (48) | 1 |
| | Para 3–4 | 26.1 (36) | 0.57 (0.31, 1.06) |
| | Grand multipara | 33.8 (22) | 0.69 (0.36, 1.30) |
| Vaginal Bleeding | No | 25.2 (101) | 1 |
| | Yes | 31.6 (6) | 1.24 (0.62, 2.47) |
| Number of Abortion | None | 24.9 (81) | 1 |
| | One | 24.7 (19) | 1.00 (0.64, 1.54) |
| | Two or more | 38.9 (7) | 1.56 (0.84, 2.86) |
| Received Fansidar (SP) during this pregnancy | No | 28.9 (59) | 1 |
| | Yes | 22.2 (48) | 0.69 (0.44, 1.07) |
| Severe anaemia (previously diagnosed by a health worker) | No | 21.5 (103) | 1 |

*(Continued)*

**Table 3.** (Continued)

| Variables | Category | Anaemic status | |
|---|---|---|---|
| | | % (n) | uOR (95% CI) |
| | Yes | 44.4 (4) | 0.42 (0.11, 1.61) |
| Malaria (Parasitemia) Status | Positive | 44.4 (4) | 1 |
| | Negative | 25.4 (102) | 0.57(0.26, 1.20) |
| Received Iron/folic acid Supplementation | No | 31.0 (48) | 1 |
| | Yes | 22.3 (59) | **0.65 (0.41, 1.00)** |
| Are you consuming alcohol such as beer, wine, spirits or local brews? | No | 25.3 (88) | 1 |
| | Yes | 27.1 (29) | 1.10 (0.61, 1.96) |
| Are you smoking? | No | 25.3 (101) | 1 |
| | Yes | 33.3 (6) | 0.67 (0.24, 21.84) |
| **PDQS: 20 food groups** | | | |
| *1. From yesterday morning to today morning did you eat dark leafy green vegetables* | Not at all | 19.0 (27) | 1 |
| | At least once | 28.8 (80) | **0.56 (0.34, 0.91)** |
| *2. From yesterday morning to today morning did you eat cruciferous vegetables* | Not at all | 25.5 (100) | 1 |
| | At least once | 25.0 (7) | 0.96 (0.39, 2.34) |
| *3. From yesterday morning to today morning did you eat dark orange vegetables and fruits* | Not at all | 26.3 (69) | 1 |
| | At least once | 24.1 (38) | 0.87 (0.55, 1.38) |
| *4. From yesterday morning to today morning did you eat other vegetables* | Not at all | 24.9 (66) | 1 |
| | At least once | 26.5 (41) | 1.07 (0.68, 1.68) |
| *5. From yesterday morning to today morning did you eat whole citrus fruits* | Not at all | 26.2 (102) | 1 |
| | At least once | 16.7 (5) | 0.56 (0.20, 1.50) |
| *6. From yesterday morning to today morning did you eat other whole fruits* | Not at all | 27.4 (84) | 1 |
| | At least once | 20.4 (23) | 0.68 (0.40, 1.15) |
| *7. From yesterday morning to today morning did you eat legumes* | Not at all | 26.7 (69) | 1 |
| | At least once | 23.5 (38) | 0.83 (0.52, 1.31) |
| *8. From yesterday morning to today morning did you eat nuts and seeds* | Not at all | 24.4 (67) | 1 |
| | At least once | 27.6 (40) | 1.18 (0.75, 1.87) |
| *9. From yesterday morning to today morning did you eat poultry* | Not at all | 25.0 (97) | 1 |
| | At least once | 31.3 (10) | 1.35 (0.62, 2.96) |
| *10. From yesterday morning to today morning did you eat fish* | Not at all | 23.6 (65) | 1 |
| | At least once | 29.2 (42) | 1.32 (0.84, 2.08) |
| *11. From yesterday morning to today morning did you eat whole grains* | Not at all | 23.5 (76) | 1 |
| | At least once | 32.0 (31) | 0.67 (0.41, 1.11) |
| *12. From yesterday morning to today morning did you eat vegetable liquid oils* | Not at all | 39.1 (18) | 1 |
| | At least once | 23.8 (89) | **0.50 (0.26, 0.94)** |
| *13. From yesterday morning to today morning did you eat white roots and tubers* | Not at all | 27.1 (51) | 1 |
| | At least once | 24.1 (56) | 0.86 (0.55, 1.34) |
| *14. From yesterday morning to today morning did you eat red meats* | Not at all | 26.8 (83) | 1 |
| | At least once | 21.8 (24) | 0.76 (0.45, 1.29) |
| *15. From yesterday morning to today morning did you eat processed meat* | Not at all | 25.8 (106) | 1 |
| | At least once | 11.1 (01) | 0.35 (0.04, 2.89) |
| *16. From yesterday morning to today morning did you eat rrefined gains and baked goods* | Not at all | 29.7 (22) | 1 |
| | At least once | 24.6 (85) | 0.77 (0.44, 1.35) |
| *17. From yesterday morning to today morning did you eat sugar sweetened beverages* | Not at all | 26.9 (67) | 1 |
| | At least once | 23.4 (40) | 0.82 (0.52, 1.28) |
| *18. From yesterday morning to today morning did you eat fried food* | Not at all | 26.2 (91) | 1 |
| | At least once | 21.9 (16) | 0.80 (0.43, 1.46) |

(*Continued*)

**Table 3.** (Continued)

| Variables | Category | Anaemic status | |
|---|---|---|---|
| | | % (n) | uOR (95% CI) |
| *19. From yesterday morning to today morning did you eat sweets and ice cream* | Not at all | 26.7 (93) | 1 |
| | At least once | 19.4 (14) | 0.65 (0.35, 1.23) |
| *20. From yesterday morning to today morning did you eat low fat diary* | Not at all | 25.2 (88) | 1 |
| | At least once | 26.8 (19) | 1.10 (0.61, 1.96) |

## Discussion

This study aimed to assess the prevalence of anaemia and to identify factors associated with anaemia among pregnant women attending antenatal clinics in the Mbeya Region of Tanzania. In the present study, 25.5% (n = 107) pregnant women were having anaemia and 74.5% (n = 313) were normal. According to the WHO classification, the prevalence of anaemia among pregnant women in this study indicates a moderate public health problem [25]. Hence, the need for effective interventions of creating nutritional awareness among these target populations.

The proportion of anaemia found in our study (25.5%) suggests that prevalence is lower than the national prevalence (57%) [8, 13]. This may be due to the improvement in antenatal health services offered over recent years. Also, the differences might be attributed by differences in regional socio-economic circumstances, cultural practices, dietary patterns, preventive health practices and diagnostic tests and improvement in attendance of antenatal care services from 43% in 2010 to 51% in 2016 [13]. Furthermore, the improvement might be

**Table 4. Multivariate logistic regression of anaemia status with socio-demographic, reproductive health, and nutrition characteristics among pregnant women attending Antenatal Clinic in Mbeya, Tanzania (n = 420).**

| Variable | Category | Anaemia status |
|---|---|---|
| | | aOR (95% CI) |
| Education level | Informal education | 1 |
| | Primary education | 0.55 (0.25,1.18) |
| | Secondary and above | 0.46 (0.18, 1.17) |
| Councils | Chunya district council | 1 |
| | Mbeya district council | **0.28 (0.11, 0.72)** |
| | Mbarali district council | 1.49 (0.66, 3.35) |
| | Kyela district council | 1.24 (0.50, 3.09) |
| | Rungwe district council | 0.90 (0.37, 2.19) |
| | Busekelo district council | 0.42 (0.12, 1.41) |
| | Mbeya city | **0.16 (0.03, 0.82)** |
| Household wealth index | Highest quintile- richest | 1 |
| | Richer | 0.74 (0.29, 1.87) |
| | Middle | 2.27 (0.96, 5.40) |
| | Poorer | **2.40 (1.05, 5.48)** |
| | Lowest quintile- poorest | 1.35 (0.81, 2.26) |
| Frequency of eating dark leafy green vegetables at least once a day | Not at all | 1 |
| | At least once | **0.53 (0.30, 0.94)** |
| Frequency of eating vegetable cooking liquid oils at least once a day | Not at all | 1 |
| | At least once | **0.56 (0.34, 0.98)** |
| Received Iron/folic acid Supplementation | No | 1 |
| | Yes | 0.77 (0.27, 2.17) |

attributed to an increase in the use of iron and folic acid supplementation for 90 days in 21.4% TDHS -MIS 2015/16 and 28.5% TNNS 2018 [8, 13, 25]. Strengthened malaria and helminths prophylaxis by the provision of antimalaria, anti-helminths and mosquito treated nets might also contribute to this improvement. Prevalence of anaemia was found to be lower in our study compared to other studies conducted in RCH clinics in Tanzania [26] and Kenya [27] that recorded higher prevalence of 47% and 57% respectively.

Anaemia in pregnant women still persists in Tanzania in spite of various national programs exist since decades [8, 13, 28]. Generally, these findings are consistent with the body of literature indicating that pregnancy increases the demand for iron, not met by food alone [29]. In line with this, the WHO recommends oral daily iron and folic acid supplementation for pregnant women [2].

In our multivariate analysis when controlling for cofounders, level of education and, iron and folic acid supplementation were not statistic significantly associated with anaemia among pregnant women in Mbeya region of Tanzania [13]. Socio-economic status that was represented by household wealth index was significantly associated with anaemia. Pregnant women in the low socio-economic status, poor wealth quintiles, were two times more likely to develop anaemia. These results could be explained by the difficulty of women in lower wealth quintiles to purchase nutritious foods compared to those in the highest wealth quintiles. Highest wealth quintile (richest) seems to have protective effect to anaemia and, this findings are in consistent with studies conducted in Northern part of Tanzania and Ethiopia where educated pregnant women had better income and eat nutritious food and hence do not get nutritional anaemia [30–32]. Moreover, these results suggest that household income independently influences anaemia among pregnant women in Mbeya Tanzania which are concordant with other similar studies conducted in Ethiopia [28, 33], Uganda [14] and Nigeria [29, 34]. Other elucidations for the association between household wealth status and anaemia include a lack of knowledge around health, as was found in a study conducted in Ghana [35], and similarly in Uganda, where a lack of health education lead to low uptake and utilization of public health interventions to combat anaemia in pregnancy [36].

The present study revealed a comprehensive picture of dietary quality among pregnant women in Mbeya region of Tanzania. Only two components of PDQS were significantly associated with anaemia in pregnant women, in multivariate analysis: eating dark green leafy vegetables and consumption of vegetable oils were strongly associated with protection from anaemia. These findings are comparable with studies conducted in the Morogoro and Dodoma Regions of Tanzania, which found that women who consumed dark green leafy vegetables resulted had higher hemoglobin and overall iron status [37], as well as with research conducted in Cameroun which found that consumption of vegetables and dark green leafy vegetables were significantly associated with lower prevalence of anaemia in pregnant women [38]. It is evident that these findings call for national programs to enhance the knowledge and skills of pregnant women in terms of use of green leafy vegetables, methods of cooking, processing of greens and growing kitchen garden.

## Strength and weaknesses of the study

This is the first study to report the prevalence of anaemia and its predictors among pregnant women (less than 28 weeks of gestation) attending antenatal clinics in the Mbeya Region of Tanzania. Despite this, the study is not without limitations. First, the study uses a cross sectional design and therefore cannot reveal causal links between anaemia and risk factors. In addition, factors such as inherited or acquired disorders that can affect hemoglobin or red blood cell synthesis were not included in this study.

## Conclusion

About quarter of the pregnant women attending antenatal services in Mbeya Region have anaemia and, that anaemia in pregnancy can be classified as moderate public health problem according to the WHO classification. Among the socio-demographic, reproductive health and nutrition characteristics of the pregnant women, socio-economic status, geographical location, consumption of green vegetables and vegetables liquid cooking oil at least once per day were identified as predictors of anaemia.

Henceforth, to combat the problem of anaemia among pregnant women there is a need to develop interventions that will strengthen health education and empower women to have a stable income. In addition, the findings indicate that there is a need to have interventions that will integrate health and nutrition education to emphasize the importance of early booking, healthy eating habits and appropriate use of Iron and folic acid supplements. Further research in cohort design is needed to explore the relationship between anaemia and the appropriate consumption of green leaf vegetable and vegetable liquid cooking oil.

## Acknowledgments

We are grateful to pregnant women, health care workers and all of those with whom we had the pleasure to work during this project.

## Author Contributions

**Conceptualization:** Fatma Abdallah, Sauli E. John, Abraham Sanga, Ramadhan Noor, Fatoumata Lankoande, Kudakwashe Chimanya, Ray M. Masumo, Germana H. Leyna.

**Formal analysis:** Fatma Abdallah, Sauli E. John, Adam Hancy, Heavenlight A. Paulo, Ramadhan Noor, Kudakwashe Chimanya, Ray M. Masumo, Germana H. Leyna.

**Funding acquisition:** Kudakwashe Chimanya.

**Methodology:** Fatma Abdallah, Sauli E. John, Adam Hancy, Abraham Sanga, Ray M. Masumo, Germana H. Leyna.

**Project administration:** Fatma Abdallah, Sauli E. John, Ramadhan Noor, Fatoumata Lankoande, Kudakwashe Chimanya, Germana H. Leyna.

**Supervision:** Ray M. Masumo, Germana H. Leyna.

**Validation:** Abraham Sanga, Ramadhan Noor, Fatoumata Lankoande, Kudakwashe Chimanya.

**Writing – original draft:** Fatma Abdallah, Sauli E. John, Adam Hancy, Abraham Sanga, Ramadhan Noor, Fatoumata Lankoande, Ray M. Masumo, Germana H. Leyna.

**Writing – review & editing:** Fatma Abdallah, Sauli E. John, Ray M. Masumo, Germana H. Leyna.

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
