## [Decision Letter · Decision Letter 0]

10 Jan 2022

PGPH-D-21-00938

Prevalence and factors associated with anemia among pregnant women attending reproductive and child health clinics in Mbeya region, Tanzania

Dear Dr. Masumo,

Thank you for submitting your manuscript to PLOS Global Public Health. After careful consideration, we feel that it has merit but does not fully meet PLOS Global Public Health’s publication criteria as it currently stands. Therefore, we invite you to submit a revised version of the manuscript that addresses the points raised during the review process.

Additional Editor Comments:

Please include recent national and international data on prevalence and risk factors for anemia if they are available. The study rationale should be more clearly stated in the introduction. Consider adding a section detailing how the questionnaire was designed or at minimum provide an overview of the study variables including details about the diet or food intake ones especially (but the other major ones as well). If the authors have the sample broken down by health care facility please include this in the text or table. There are some typographic errors that the reviewers have detailed require correction (in the manuscript abstract, main text and tables). Please check to ensure the variable categories and spellings are appropriate. Are the wealth quintiles in the correct order, and if so, how do the authors explain the results of greater anemia with higher wealth? Include the results of the bi-variate assessments in the table (unadjusted ORs). Further discussion of underlying factors that might be responsible for the study findings and the implications of the findings are required.

We look forward to receiving your revised manuscript.

Kind regards,

Colleen M. Davison

Academic Editor

Journal Requirements:

1. Please amend your detailed Financial Disclosure statement. This is published with the article, therefore should be completed in full sentences and contain the exact wording you wish to be published.

ii). State the initials, alongside each funding source, of each author to receive each grant.

iii). State what role the funders took in the study. If the funders had no role in your study, please state: “The funders had no role in study design, data collection and analysis, decision to publish, or preparation of the manuscript.”

2. Please include the Funding Information in the system. This should have the same information of Financial Disclosure statement.

3. Please amend your Data Availability Statement and indicate where the data may be found

Reviewers' comments:

Reviewer's Responses to Questions

**Comments to the Author**

1. Does this manuscript meet PLOS Global Public Health’s publication criteria? Is the manuscript technically sound, and do the data support the conclusions? The manuscript must describe methodologically and ethically rigorous research with conclusions that are appropriately drawn based on the data presented.

Reviewer #1: Yes

Reviewer #2: Yes

2. Has the statistical analysis been performed appropriately and rigorously?

Reviewer #1: Yes

Reviewer #2: Yes

3. Have the authors made all data underlying the findings in their manuscript fully available (please refer to the Data Availability Statement at the start of the manuscript PDF file)?

Reviewer #1: No

Reviewer #2: Yes

4. Is the manuscript presented in an intelligible fashion and written in standard English?

Reviewer #1: Yes

Reviewer #2: Yes

5. Review Comments to the Author

Reviewer #1: Introduction

1. Introduction need to improve with recent data both nationally and internationally.

2. Authors needs to highlight the rational of the study clearly.

Methodology

1. Data collection procedure needs to explain clearly.

2. authors can add a section named Questionnaire design to explain the questionnaire part.

3. in data analysis section, clear outcome and predictor variable.

Discussion

1.The discussion generally situates the present study within the broader literature in the study field, which is fine. What it falls short is that it does not really interpret the findings as to the possible underlying factors responsible for the study findings, the inferences from and the implications of the findings. To me, the discussion in its current form mainly compares, and rarely interprets, infers and implies.

2.The discussion of the study findings should be strengthened. Authors should do well to assign reasons for their findings. Authors should extensively discuss the implications of the study in terms of policy, practice and future research.

Other comments: This manuscript would benefit from language editing.

Reviewer #2: Abdallah et al. have presented an interesting study on the prevalence and factors associated with anemia among pregnant women attending reproductive and child health clinics in Mbeya region, Tanzania. The study relies on the data from hospital-based surveys and reports that pregnant women with second and middle wealth quintiles were two times more likely to be anemic. The authors also have a well-written method section, especially the sampling part of data and process involved in the collection, and this study addresses some key questions. However, it will be great if the authors can have more clarity for the following points:

I did not find the sample for every RCH facility, which means how many samples were collected from each RCH. The authors give the collected sample distribution by RCH in the appendix. That will be helpful to understand the cluster of anemic women because, in the introduction part, it mentions that geography also plays a role in anemia that means unavailability of health facilities or other reasons. Here I did not understand these facilities are public or private entities give some explanation.

What are food information had been collected is missing in the method section. As in the introduction section, it is mentioned that diet has an essential role in anemia, so the diet group must be in the study.

Further, what criteria of prime diet quality score are not come out.

Is PDQS similar to dietary diversity?

Why have authors categorized PQDS in only two groups?

Control variables were missing in the methods section include it by which the reader can quickly come out the variable descriptions.

The information of outcome variables was also missing.

In the result section (line no 6), the sentence “Out of the 240 respondents to the questionnaire,” is it 420 or 240” seems confusing.

In table 1 and 2 need some modification in the categories of the following variables

(a) The age group should be in four categories (15-19, 20-24, 25-29, and ≥30 years).

(b) Marital status should be in three categories (Married, Cohabit, Single/Divorced)

(c) Anemia status (no, 10.0- 10.9 g/dl =mild, <7.0 – 9.9 g/dl= moderate) define in the variable’s description mild and moderate only. If it is binary, then keep in yes and no only.

In the result section “Prevalence of anemia in the study area,”; I have not found the table regarding the prevalence of anemia. Is it any supplementary tables? I also did not find the supplementary table. So, make it clear.

Further, the bivariate model is missing from the analysis, which is essential to understand the association between outcome and independent variables; keep it in table 1.

The author has mentioned that they have collected 21 food lists, but I did not find the dietary diversity-related variable; I think the authors should include it.

Univariate and multivariate findings indicate that pregnant women from the second and middle wealth quintiles have a higher chance of anemic than lower wealth quintiles. That means as wealth quintiles increase, odds of anemia increase. Why it is happening, give some explanation or check it.

Life style-related variables like consumption of alcohol, smoking are missing; if authors have collected, then include them in the analysis.

During interpretation of the result no need to give the p-value; CI is sufficient.

The multivariate analysis includes all control variables instead of only selected variables.

In the discussion section, there is no need to repeat all the findings; only explain in terms of higher or lower association. Add if any government policy is going on to improve the nutritional status of women.

Anemia is modifiable to explain the finding make it cautious. Further sample study only based on those women who were taking the health facility means they have some previous knowledge about health, hence write these are the terms in the limitation.

Please check the abstract result; the author has mentioned “less associated,” I think it will be higher or more likely. Similarly, check the spelling of variables in table 1.

6. PLOS authors have the option to publish the peer review history of their article (what does this mean?). If published, this will include your full peer review and any attached files.

**Do you want your identity to be public for this peer review?** For information about this choice, including consent withdrawal, please see our Privacy Policy.

Reviewer #1: No

Reviewer #2: No

---

## [Decision Letter · Decision Letter 1]

15 Mar 2022

PGPH-D-21-00938R1

Prevalence and factors associated with anemia among pregnant women attending reproductive and child health clinics in Mbeya region, Tanzania

Dear Dr. Masumo,

Thank you for submitting your manuscript to PLOS Global Public Health. After careful consideration, we feel that it has merit but does not fully meet PLOS Global Public Health’s publication criteria as it currently stands. Therefore, we invite you to submit a revised version of the manuscript that addresses the points raised during the review process.

EDITOR: Thank you for your revised manuscript. We have determined a few remaining edits that should still be addressed:

1. Please use the same spelling of “anaemia” throughout the study

2. In the revised manuscript line number 141, keep OR in brackets.

3. Inline number 143, the author has used the “near” word. I think it would be “equal”.

4. No need to show the odds values in the discussion section.

5. Please do not include any statistics in the conclusion section.

6. In Table 2, the malaria variables sample is only 411, but I have read it was 420. So, please check it for all variables.

7. In Table 2, the percentage values are exceeding 100 percent, so please check it and correct it.

8. In Table 3, the not employed column total percentage is only 80% make sure it will 100%.

9. In table 3, keep only “anaemic column”, no need for “not anaemic column”

Reviewers' comments:

Reviewer's Responses to Questions

**Comments to the Author**

1. If the authors have adequately addressed your comments raised in a previous round of review and you feel that this manuscript is now acceptable for publication, you may indicate that here to bypass the “Comments to the Author” section, enter your conflict of interest statement in the “Confidential to Editor” section, and submit your "Accept" recommendation.

Reviewer #2: All comments have been addressed

2. Does this manuscript meet PLOS Global Public Health’s publication criteria? Is the manuscript technically sound, and do the data support the conclusions? The manuscript must describe methodologically and ethically rigorous research with conclusions that are appropriately drawn based on the data presented.

Please ensure that your decision is justified on PLOS Global Public Health’s publication criteria and not, for example, on novelty or perceived impact.

We look forward to receiving your revised manuscript.

Kind regards,

Colleen M. Davison

Academic Editor

Journal Requirements:

1. Please update your Competing Interests statement. If you have no competing interests to declare, please state: “The authors have declared that no competing interests exist.”

Reviewers' comments:

Reviewer's Responses to Questions

**Comments to the Author**

1. If the authors have adequately addressed your comments raised in a previous round of review and you feel that this manuscript is now acceptable for publication, you may indicate that here to bypass the “Comments to the Author” section, enter your conflict of interest statement in the “Confidential to Editor” section, and submit your "Accept" recommendation.

Reviewer #2: All comments have been addressed

2. Does this manuscript meet PLOS Global Public Health’s publication criteria? Is the manuscript technically sound, and do the data support the conclusions? The manuscript must describe methodologically and ethically rigorous research with conclusions that are appropriately drawn based on the data presented.

Reviewer #2: Yes

3. Has the statistical analysis been performed appropriately and rigorously?

Reviewer #2: Yes

4. Have the authors made all data underlying the findings in their manuscript fully available (please refer to the Data Availability Statement at the start of the manuscript PDF file)?

Reviewer #2: Yes

5. Is the manuscript presented in an intelligible fashion and written in standard English?

Reviewer #2: Yes

6. Review Comments to the Author

Reviewer #2: I really appreciate author for incorporating the comments. I have some general comments

1. Please use the same spelling of “anaemia” throughout the study

2. In the revised manuscript line number 141, keep OR in brackets.

3. Inline number 143, the author has used the “near” word. I think it would be “equal”.

4. No need to show the odds values in the discussion section.

5. Please do not include any statistics in the conclusion section.

6. In Table 2, the malaria variables sample is only 411, but I have read it was 420. So, please check it for all variables.

7. In Table 2, the percentage values are exceeding 100 percent, so please check it and correct it.

8. In Table 3, the not employed column total percentage is only 80% make sure it will 100%.

9. In table 3, keep only “anaemic column”, no need for “not anaemic column”

7. PLOS authors have the option to publish the peer review history of their article (what does this mean?). If published, this will include your full peer review and any attached files.

**Do you want your identity to be public for this peer review?** For information about this choice, including consent withdrawal, please see our Privacy Policy.

Reviewer #2: No

---

## [Decision Letter · Decision Letter 2]

20 Jul 2022

PGPH-D-21-00938R2

Prevalence and factors associated with anaemia among pregnant women attending reproductive and child health clinics in Mbeya region, Tanzania

Dear Dr. Masumo,

Thank you for submitting your manuscript to PLOS Global Public Health. After careful consideration, we feel that it has merit but does not fully meet PLOS Global Public Health’s publication criteria as it currently stands. Therefore, we invite you to submit a revised version of the manuscript that addresses the points raised during the review process.

Thank you for submitting your manuscript to PGPH for publication. It has been reviewed by three independent peer reviewers and found to have merit. However, they have identified some issues (attached/appended) that you need to address to improve the manuscript. 

After going through your manuscript, I have also identified some shortfalls which need to be addressed before it can be accepted for publication:

1. It is OK to write aOR (adjusted Odds Ratio) after spelling it out at first use. Effect these changes throughout the manuscript

2. Why did you use a power of 95% instead of 80% (a commonly used parameter).

3. Is the 1.5 the same as design effect? clarify

4. Is the 420 stated in the preceding line inclusive of non-response rate of 10% in the second line. This appears a bit confusing.

5. Is the assumption of 0.10 as the difference between the two groups the same as intracluster correlation coefficient (ICC)?

6. The study used two stage sampling strategy. This suggests that design effect must be accounted for in all the analysis. It appears this was not done. However, if it was accounted for, explain how it was done. And if it was not done, explain why.

7. The ethical statement needs a bit more details, especially regarding the consenting process e.g. Was the risks and benefits of the study explained to the participants? Were they informed of their rights to refuse to participate in the study or withdraw from it at any time, without consequences? What about the issues of confidentiality? etc.

8. Remove all p-values, ORs and CIs from the discussion section

We look forward to receiving your revised manuscript.

Kind regards,

Dickson Abanimi Amugsi, PhD

Academic Editor

Journal Requirements:

Additional Editor Comments (if provided):

Reviewers' comments:

Reviewer's Responses to Questions

**Comments to the Author**

1. If the authors have adequately addressed your comments raised in a previous round of review and you feel that this manuscript is now acceptable for publication, you may indicate that here to bypass the “Comments to the Author” section, enter your conflict of interest statement in the “Confidential to Editor” section, and submit your "Accept" recommendation.

Reviewer #3: All comments have been addressed

Reviewer #4: (No Response)

Reviewer #5: All comments have been addressed

2. Does this manuscript meet PLOS Global Public Health’s publication criteria? Is the manuscript technically sound, and do the data support the conclusions? The manuscript must describe methodologically and ethically rigorous research with conclusions that are appropriately drawn based on the data presented.

Reviewer #3: Yes

Reviewer #4: Partly

Reviewer #5: Yes

3. Has the statistical analysis been performed appropriately and rigorously?

Reviewer #3: I don't know

Reviewer #4: No

Reviewer #5: Yes

4. Have the authors made all data underlying the findings in their manuscript fully available (please refer to the Data Availability Statement at the start of the manuscript PDF file)?

Reviewer #3: Yes

Reviewer #4: Yes

Reviewer #5: Yes

5. Is the manuscript presented in an intelligible fashion and written in standard English?

Reviewer #3: Yes

Reviewer #4: Yes

Reviewer #5: No

6. Review Comments to the Author

Reviewer #3: The authors have clarified most of the questions raised in the previous reviews and have adjusted the methodology and discussion portion. The manuscript does a good job highlighting the prevalence of anemia and the factors associated with it in pregnant women, highlighting an important health issue from an underserved setting.

We have a couple of remaining comments:

Line 29-30; The sentence can be improved by rewording. Suggesting: During pregnancy, there is a marked increase in the minimum adult requirement by almost 2-3 fold for iron and 10-20 folds for folate.

Line 88-89: Please mention the criteria for eligibility

Line 232- Instead of despite of say “in spite of”

Line 233-238- No need to repeat results in discussion. Delete this : In our study (univariate level), anaemia was significantly associated with pregnant women who had attended a secondary education or higher, household wealth index (in poor and middle quintiles), received a supplementation of iron and folic acid, geographical location (Mbeya district council, Busekelo district council and Mbeya city council), as well as with the consumption of dark leafy green vegetables and, vegetable liquid oil.

Line 238-239- The sentence Generally, these findings are in consistent with the body of literatures that indicate that pregnancy increases the demand for iron, a demand that is often not met by food alone is confusing. Rephrase: Generally, these findings are consistent with the body of literature indicating that pregnancy increases the demand for iron, not met by food alone

Line 272-273- Rephrase: This is the first study to report the prevalence of anaemia and its predictors among pregnant women (less than 28 weeks of gestation)

attending antenatal clinics in the Mbeya Region of Tanzania.

Line 277 – Instead of casual say causal

Line 288- Instead of risk factors say predictors.

Reviewer #4: They need to re-analyze the parity and anemia relationship

Reviewer #5: Dear authors,

Thank you for working on the revision of the manuscript. As newly assigned reviewer, I have minor comments that I would like the authors to address. These can be found in the manuscript directly (pdf) on pages 55 to 82.

I would like also to suggest that the authors revise the English of the manuscript.

All the best

7. PLOS authors have the option to publish the peer review history of their article (what does this mean?). If published, this will include your full peer review and any attached files.

**Do you want your identity to be public for this peer review?** For information about this choice, including consent withdrawal, please see our Privacy Policy.

Reviewer #3: No

Reviewer #4: **Yes: **Awor

Reviewer #5: **Yes: **Souheila Abbeddou

---

## [Decision Letter · Decision Letter 3]

13 Sep 2022

Prevalence and factors associated with anaemia among pregnant women attending reproductive and child health clinics in Mbeya region, Tanzania

PGPH-D-21-00938R3

Dear Dr Masumo,

We are pleased to inform you that your manuscript 'Prevalence and factors associated with anaemia among pregnant women attending reproductive and child health clinics in Mbeya region, Tanzania' has been provisionally accepted for publication in PLOS Global Public Health.

Best regards,

Dickson Abanimi Amugsi, PhD

Academic Editor

Reviewer Comments (if any, and for reference):

Reviewer's Responses to Questions

**Comments to the Author**

1. If the authors have adequately addressed your comments raised in a previous round of review and you feel that this manuscript is now acceptable for publication, you may indicate that here to bypass the “Comments to the Author” section, enter your conflict of interest statement in the “Confidential to Editor” section, and submit your "Accept" recommendation.

Reviewer #4: All comments have been addressed

2. Does this manuscript meet PLOS Global Public Health’s publication criteria? Is the manuscript technically sound, and do the data support the conclusions? The manuscript must describe methodologically and ethically rigorous research with conclusions that are appropriately drawn based on the data presented.

Reviewer #4: Yes

3. Has the statistical analysis been performed appropriately and rigorously?

Reviewer #4: Yes

4. Have the authors made all data underlying the findings in their manuscript fully available (please refer to the Data Availability Statement at the start of the manuscript PDF file)?

Reviewer #4: Yes

5. Is the manuscript presented in an intelligible fashion and written in standard English?

Reviewer #4: No

6. Review Comments to the Author

Reviewer #4: Thank you for addressing the earlier comments

7. PLOS authors have the option to publish the peer review history of their article (what does this mean?). If published, this will include your full peer review and any attached files.

**Do you want your identity to be public for this peer review?** For information about this choice, including consent withdrawal, please see our Privacy Policy.

Reviewer #4: No
